# *OsFBN6* Enhances Brown Spot Disease Resistance in Rice

**DOI:** 10.3390/plants13233302

**Published:** 2024-11-25

**Authors:** Fang-Yuan Cao, Yuting Zeng, Ah-Rim Lee, Backki Kim, Dongryung Lee, Sun-Tae Kim, Soon-Wook Kwon

**Affiliations:** 1Department of Plant Bioscience, College of Natural Resources and Life Science, Pusan National University, Miryang 50463, Republic of Korea; no.1lvtu@outlook.com (F.-Y.C.); zyt-edu@outlook.com (Y.Z.); aar5430@gmail.com (A.-R.L.); m5505@gmail.com (S.-T.K.); 2Life and Industry Convergence Research Institute, Pusan National University, Miryang 50463, Republic of Korea; uptfamily@hanmail.net (B.K.); dongryung.lee0103@gmail.com (D.L.)

**Keywords:** rice brown spot, *C. miyabeanus*, jasmonic acid, *OsFBN6*, overexpression

## Abstract

Brown spot (BS) is caused by necrotrophs fungi *Cochliobolus miyabeanus* (*C. miyabeanus*) which affects rainfed and upland production in rice, resulting in significant losses in yield and grain quality. Here, we explored the meJA treatment that leads to rice resistance to BS. Fibrillins (FBNs) family are constituents of plastoglobules in chloroplast response to biotic and abiotic stress, many research revealed that *OsFBN1* and *OsFBN5* are not only associated with the rice against disease but also with the JA pathway. The function of *FBN6* was only researched in the *Arabidopsis.* We revealed gene expression levels of *OsFBN1*, *OsFBN5*, *OsFBN6* and the JA pathway synthesis first specific enzyme *OsAOS2* following infection with *C. miyabeanus*, *OsAOS2* gene expression showed great regulation after *C. miyabeanus* and meJA treatment, indicating JA pathway response to BS resistance in rice. Three FBN gene expressions showed different significantly regulated modes in *C. miyabeanus* and meJA treatment. The haplotype analysis results showed *OsFBN1* and *OsFBN5* the diverse Haps significant with BS infection score, and the *OsFBN6* showed stronger significance (**** *p* < 0.0001). Hence, we constructed *OsFBN6* overexpression lines, which showed more resistance to BS compared to the wild type, revealing *OsFBN6* positively regulated rice resistance to BS. We developed *OsFBN6* genetic markers by haplotype analysis from 130 rice varieties according to whole-genome sequencing results, haplotype analysis, and marker development to facilitate the screening of BS-resistant varieties in rice breeding. The Caps marker developed by Chr4_30690229 can be directly applied to the breeding application of screening rice BS-resistant varieties.

## 1. Introduction

Rice brown spot (BS) caused by necrotrophs *C. miyabeanus* (Anamorph: *Bipolaris oryzae*), a fungus first described by Drechsler (1934) and Dastur (1942), belongs to the new genus *Cochliobolus* [1]. BS affects both rain-fed and upland rice production, causing losses in grain quality and yield by 6−90% [2,3,4]. It particularly occurs in environments where water supply is scarce and is often combined with imbalances in plant mineral nutrition, specifically a lack of nitrogen [5,6] and a deficit of silicon. Timely transplantation of rice (between July–August) and sufficient levels of nitrogen enhance rice growth and minimize *C. miyabeanus* infection [7]. Si is presumably not essential for plant growth and development [8]. The application of various Si sources to Si-deficient paddy soils dramatically reduces the incidence and severity of BS caused by *C. miyabeanus* [9]. Si mounts resistance to *C. miyabeanus* by preventing the fungus from hijacking the rice ethylene machinery [10]. However, applying fertilizer blindly cannot fundamentally solve BS, so this study decided to look for disease-resistant factors at the genetic level.

Plants resist the attack by microbial pathogens by regulating the production of hormones, with the key signaling molecules including salicylic acid (SA), JA, and ethylene (ET) [11]. SA, JA, and ET can activate resistance-related genes, which significantly vary in timing, quantity, and composition depending on distinct types of pathogens [12]. The hypothesized mode of interaction between the SA and JA/ET signaling pathways appears to be mutual antagonism during resistance to necrotrophic and biotrophic pathogens [13,14]. In a study on exogenous abscisic acid inducing resistance against *C. miyabeanus*, abscisic acid activated the defense response against rice pathogens by suppressing the action of ET [15]. Research into Arabidopsis against *C. miyabeanus* (strain HIH-1) showed more resistance by genetic disruption mutants of the ET and JA signaling pathways than wild-type [16]. Allene oxide synthase (AOS) (cytochrome P450 74A) is the first specific enzyme, which leads to JA synthesis in rice chloroplast [17]. *OsAOS2* activates pathogenesis-related (PR) genes and enhances resistance to *M. oryzae* [18]. JA treatment of rice has indicated some proteins associated with defense response, such as PR-1, PR-5, β-1,3-glucanase, and a receptor-like kinase [19]. *OsWRKY13* can enhance the resistance of rice plants to bacterial blight and fungal blast, and suppression of JA-synthesis-related and JA-responsive genes [20]. *OsWRKY30* enhances necrotrophic resistance against sheath blight by increasing JA accumulation and JA-responsive *PR* expression [21].

Fibrillins (FBNs) constitute a nucleus-encoded plastid-associated protein family that contains a chloroplast transport peptide (CTP) domain and a plastid lipid-associated protein (PAP), which are constituent plastoglobules in chloroplasts [22]. FBNs are involved in plant growth and development, tolerance to oxidative stress, and hormone signaling [23]. The first FBNs were observed in chloroplasts of dog rose (*Rosa canina*) and bell pepper (*Capsicum annuum*) [24]. To date, *FBN* gene families have been grouped into 11 clades in *Arabidopsis* and rice [25]. All of *FBNs* are found in the chloroplast in *Arabidopsis* and rice [25,26,27]. *FBN1* expression in tomatoes and peppers is induced by several diseases [28,29]. However, FBN1 decreases the infection level of mosaic virus in tobacco [30]. In *Arabidopsis*, FBN4 (*At3g23400*) responds to the pathogen-associated molecular pattern (PAMP) in the form of phosphorylation during the defense response to *Pseudomonas syringae* DC3000 [31], and FBN1 and FBN2 respond to high-light and low-temperature conditions by delaying anthocyanin accumulation [32]. FBN4 is related to tolerance to bacterial pathogens, herbicides, high light intensity, and ozone in apple and *Arabidopsis* [33]. In rice, *OsFBN1* overexpression reduces the grain-filling rate and JA levels under heat stress. The FBN5 has been found to have the same function in rice and *Arabidopsis*, lack of *OsFBN5* in rice results in increased susceptibility to light and cold stress and suppresses the expression of JA-related genes to ≤10% compared to that in the wild type [34,35]. The research into FBN6 shows that FBN6 is located in the thylakoid and envelope membranes in *Arabidopsis* chloroplast [36]. An *FBN6*-deficient mutant in *Arabidopsis* has been reported to exhibit decreased carotenoid production during the seedling de-etiolation period under high-light conditions [37].

Based on these findings, we aimed to explore the gene expression level of *OsFBN1*, *OsFBN5* and *OsFBN6* following infection with *C. miyabeanus* and meJA treatment. The *OsFBN6* resistance to *C. miyabeanus* was revealed by constructing overexpressed lines and developing markers by haplotype analysis for the identification of future resistant rice varieties

## 2. Results

### 2.1. Phenotype and Genes Expression Analysis

To verify the expression patterns of *OsFBN6*, *OsFBN1*, and *OsFBN5*, we examined both resistant (R) and susceptible (S) rice lines following infection with *C. miyabeanus* (Figure 1A). The R line was still resistant to BS after treatment with meJA, and the BS lesions in the meJA-treated S line became relatively small, showing significant resistance to BS compared to that without MeJA treatment (Figure 1B). *OsFBN6* expression significantly increased after infection in the R line (*p* < 0.01), whereas it significantly decreased by meJA treatment in both the R and S lines (Figure 1C). *OsFBN1* expression decreased after infection in both R and S lines and decreased with meJA treatment in the S line (Figure 1D). *OsFBN5* expression decreased after infection in the R line but increased in the S line, and the expression decreased in both lines by meJA treatment (Figure 1E). To verify whether rice-disease-related genes known in the JA pathway are also associated with BS resistance, gene expression results revealed *OsWRKY13* showed a reverse trend with the JA-synthesis-related gene *OsAOS2* after *C. miyabeanus* infection, the opposite expression trend is shown after meJA treatment in R and S lines (Appendix A), indicating that *OsWRKY13* is not only involved in against sheath blight and blast but also responds to *C. miyabeanus* in the JA pathway. The *OsWRKY30* expression trend has no difference after *C. miyabeanus* and meJA treatment (Appendix A), indicating that *OsWRKY30* does not participate against BS in rice plants. *OsAOS2* and *OsFBN6* gene expression levels of R and S lines both increased after BS infection and decreased after meJA treatment. *OsFBN6* exhibited expression patterns similar to that of *OsAOS2*. The different aspect is the R line of *OsFBN6* showed significantly increased expression than the S line, but the S line of *OsAOS2* showed an over 10-fold expression increase than the R line after BS infection.

### 2.2. Haplotype Analysis

To reveal the effects of genotype differences on phenotype in 130 rice varieties, we carried out haplotype analysis for three FBN genes. Duncan’s test was used to analyze the average values of the four haplotypes. There were three SNPs in the introns revealed by haplotype analysis of *OsFBN6*, one in the coding sequence, and three in the promoter region. The reference sequences in all haplotype analyses are from NIP. The results showed a significant difference in infection average score between haplotypes 2 and 3, and haplotypes 1 and 2 showed significantly more susceptibility than haplotypes 3 and 4 with significant differences at α = 0.01. The SNP mutant A to C in Chr4_30688750 was found in haplotype 2 (Figure 2A), which is located in CDS region 228 from the ATG and lets GCU mutate into GCG (reverse strand). The results of amino acid sequence alignment showed that the amino acid still did not change (Appendix A). According to the haplotype network analysis of each subspecies, haplotypes 1 and 2 were both temperate japonicas, and haplotype 3 was almost entirely indica (Figure 2B). Duncan’s test results showed significant distinctions between each haplotype (*p* < 0.0001). Haplotype analysis of *OsFBN1* classified six haplotypes. Haplotypes 5 and 6 showed the lowest infection average score than the other haplotypes. Six SNPs were identified in haplotype 5, and the SNP Chr09_2548752 was identified in the promoter region of haplotype 6 (Appendix A). Duncan’s test results showed significant distinctions between each haplotype (*p* = 0.0073). Haplotype network analysis of each subspecies showed that haplotype 5 contained aus and indica, and haplotype 6 was constituted only of indica (Appendix A). Haplotype analysis of *OsFBN5* classified five haplotypes, and haplotype 5 had the lowest average infection score than those of the other haplotypes. The only classified SNP was Chr04_20872801 in the promoter region (Appendix A). Duncan’s test results showed significant distinctions between each haplotype (*p* = 0.0008). Haplotype network analysis showed that haplotype 5 was entirely composed of indica and was more resistant to BS than haplotype 3, which was also composed of indica, because of the SNP Chr04_20872801 in the former haplotype (Appendix A). The violin plots of each haplotype of OsFBN6 show that the median BS infection and fraction distribution of hap 2 and 3 are much lower than that of hap 1 and 4 (Figure 2C). The BS infection score distribution in each haplotype of *OsFBN1* and *OsFBN5* median values of haplotypes from hap 3 onwards were significantly lower than those of hap1 and 2, and the number of varieties is only a minority (Figure 2D,E). The number of varieties contained in hap 2 and 3 haplotypes is about half the number of all haplotypes in *OsFBN6*, which is suitable for developing markers to screen for BS-resistant varieties.

### 2.3. Marker Development and Verification by SNPs

To identify the four haplotypes of *OsFBN6* in more rice varieties so that BS-resistant varieties can be used in breeding, we selected three SNPs as markers based on the haplotype analysis (Figure 3A). Two of them used the Arms primer, and the last used the Caps primer, which restricts the enzyme cutting site to SspI (Figure 3B). The marker of Chr4_30688750 showed mutant allele C in hap 2; the marker of Chr4_30688916 showed wild allele A between hap 1 and 4; the marker of Chr4_30690229 showed mutant allele T in hap 4. According to the three marker design results, four haplotypes were identified in different varieties. We selected one variety from each haplotype to extract DNA and obtained the same results through PCR analysis. The second fragment of Chr4_30688750 was 190 bp, indicating that the allele is mutant C and can be used to identify hap 2; Chr4_30688916 showed wild allele A (186 bp) and mutant allele C (217 bp) by identifying haps 1 and 2 and haps 3 and 4, respectively (Figure 4A). The last marker, Chr4_30690229, was used to distinguish between hap 3 and hap 4, and the mutant allele T was cut by SspI; thus, hap 3 obtained two small fragments of 380 bp and 328 bp (Figure 4B). These results showed that the developed markers could be used to identify BS-resistant varieties.

### 2.4. Analysis of Overexpressed Lines

We generated a CaMV 35S promoter-containing transgenic NIP T_0_ and screened overexpressing plants through rice leaf RT-qPCR for *OsFBN6* (Figure 5A). Transgenic T_1_ lines overexpressing *OsFBN6* (*OsFBN6*-OEs) showed more *C. miyabeanus* resistance than NIP, and leaf conditions were greener than that in NIP (Figure 5B). The OE-1 and OE-5 lines showed significantly lower levels of BS disease because *OsFBN6* expression was >2-fold that of the NIP (Figure 5C). To analyze phenotypic differences more directly, we calculated the lesion-relative area for each leaf and demonstrated that both the infection numbers and lesion areas of the OE lines indicated higher resistance to *C. miyabeanus* compared to NIP (Figure 5D). We compared the total infection of more than three leaves on the macro data level, and the total lesion-relative area of the OE lines was significantly lower than that of the NIP (Figure 5E).

## 3. Methods

### 3.1. Plant Materials and Whole Genotype Collection

This study selected Korean native resistant (R) (Jejubukjeju-2002-171) and susceptible (S) (Muando) rice varieties according to different infection levels from 130 rice cultivars that included 62 temperate japonica, 19 tropical japonica, 41 indica, and 8 aus varieties (Appendix A). The 130 rice germplasms were sequenced using an Illumina HiSeq 2500 Sequencing System (Illumina Inc., San Diego, CA, USA) [38]. The Genome-wide resequencing of 130 rice cultivars data has been published online DOI: 10.1186/s12864-016-2734-y, and reference high-quality SNPs in cultivars from chromosomes 1–12 within additional files 4 and 6. The average genome coverage was 8×, and the filtered reads were aligned to the rice reference genome (IRGSP 1.0) https://ensembl.gramene.org/Oryza_sativa/Info/Index?db=core;r=4:18719520-19147493 (accessed on 21 November 2024). The following parameters were used to filter genotypes for genome-wide association studies, such as minor allele frequency > 1%, missing data < 1%, and heterozygosis ratio < 5%, and these were implemented using Plink v. 1.9 software [39]. Finally, approximately 1.4 million single nucleotide polymorphisms (SNPs) were selected from the 6.5 million raw SNP database.

### 3.2. Evaluating Pathogen Inoculation and Methyl Jasmonate (meJA) Treatment

The *C. miyabeanus* 36 strain was obtained from the Korean GenBank of the RDA. The strain was grown for sporulation on potato dextrose agar at 28 °C under light stress [15] to observe the morphology and microscopically quantity spores after approximately 2−3 weeks (Appendix A). One day before infection, 0.1 mM meJA with 0.02% (*v*/*v*) Tween 20 was spread on rice leaves (three plants/2 mL). The conidia of *C. miyabeanus* 36 (Appendix A) were collected from cultures using double distilled H_2_O containing 0.02% (*v*/*v*) Tween 20, and the concentration was adjusted to 1 × 10^5^ spores/mL. For inoculation, 3-week-old plants (seedling-leaf stage) planted in 96-well PCR plates and cultivated in a hydroponics system were inoculated by spraying with conidial suspension until all leaves were covered with fine droplets. The inoculated plants were kept in a dark humidity chamber for 24 h at 28 ± 1 °C to facilitate fungal penetration, and then transferred to greenhouse conditions (28 ± 1 °C, 16:8 h light–dark cycle) for disease development. Disease severity was evaluated at 7 days post-infection, and the degree of infection was distinguished between R and S [40]. All infection assays were repeated at least thrice.

### 3.3. Gene Expression Analysis by Using Real-Time Quantitative Polymerase Chain Reaction (RT–qPCR)

Gene expression was assessed in the R and S rice varieties. Total RNA was extracted from seedling leaves after 7 days post-infection using a Takara MiniBEST Plant RNA Extraction Kit (Takara Bio Inc., Shiga 525-0058, Japan), and the samples were treated with RNase-free DNase (QIAGEN 19300 Germantown Rd., Germantown, MD 20874, USA) to remove genomic DNA. Complementary DNA was performed on 2 μg of total RNA using a SuperScript III Kit (Thermo Fisher Scientific Inc., Carlsbad, CA, USA) with primers for the target genes designed using the NCBI Primer-Basic Local Alignment Search Tool (Appendix A). RT-qPCR was performed with the FastStart™ Universal SYBR^®^ Green Master (Rox) (Roche, Germany) and QuantStudio™ 1 Real-Time PCR Instrument (Marsiling, Singapore 739256) to achieve results in the two-step thermal cycler program, using standard curve-based quantification with 60 °C annealing temperature and three technical replicates [41]. The cycle number threshold (Ct value) was employed to calculate relative mRNA expression levels. The Ct value for each target gene was normalized by subtracting the Ct value of the rice (ACTIN 1, LOC_Os03g50885) gene. The −2^ΔΔCt^ method was used to calculate relative gene expression [42]. The differences between the expression level average arrays were measured using the Student’s *t*-test by the IBM SPSS Statistics 26 software.

### 3.4. Haplotype Analysis and Prediction of Protein Structure

All SNP markers from genes and promoters were used. The average score and variety count were determined from the phenotypic data for each rice subspecies, and haplotypes that were significantly associated with the phenotype were identified. An online tool, Gene Structure Display Server v.2.0 [43], was used to visualize gene structure and SNP positions. Haplotype network analysis was performed using PopART v.1.7 [44]. The differences between haplotype infection average arrays were analyzed using *t*-test and Duncan’s test using IBM SPSS Statistics v.26 software. Amino acid sequences were aligned using BioEdit v. 7.2 software.

### 3.5. Marker Development

To apply dominance haploidy to breeding, two methods were used for developing markers for screening dominant varieties: the amplification refractory mutation system (ARMS) [45] using the tetra-primer ARMS-PCR online tool (http://primer1.soton.ac.uk/primer1.html) (accessed on 21 November 2024) and the cleaved amplified polymorphic sequence (CAPS) method [46] using SnapGene v. 8.0 software.

### 3.6. Phenotype of OsFBN6-Overexpressing Lines

The gateway cloning method has been used to construct the overexpression plasmid by the Thermofisher online protocol [47] (https://assets.thermofisher.com/TFS-Assets/LSG/manuals/gateway_clonaseii_man.pdf) (accessed on 21 November 2024). The cloning plasmid was constructed by vector pDONR21 and transformed into *Escherichia coli* DH5α, and the expression plasmid was constructed by vector pGWB517 and then transformed to Agrobacterial LBA4404. Nipponbare (NIP) callus was infected via Agrobacterium transformation. The seeds of 10 T0 lines were collected, and each line was planted with 20 seeds to grow three weeks old for BS infection. The step of pathogen inoculation by *C. miyabeanus* was the same as in 2.1. We collected all plants for the identification of overexpressed plants using RT–qPCR after we evaluated the phenotype.

## 4. Discussion

Initially, genetic markers were used to determine the order of genes along chromosomes in genetic mapping [48]. The utilization of molecular markers has increased the speed and accuracy of plant genetic analysis and promoted the efficiency of crop breeding [49]. Three major groups of allelic variations can be identified within the genome of the same species: simple sequence repeats, segmental insertions/deletions, and SNPs [50,51,52]. As nucleotide differences between individuals and every SNP in a single copy of DNA are potentially useful markers, their numbers are virtually unlimited [53]. In *Arabidopsis thaliana, ACCELERATED CELL DEATH 6* (*ACD6*) has demonstrated allelic diversity at a single locus, and compared with the reference allele, it has increased resistance to five pathogens from different phyla [54]. Haplotype analysis is an important aspect of rice breeding. When some genomic haplotypes are known, they can be used as alleles for single multiallelic markers [55]. Using whole-genome sequencing results from 130 rice varieties, SNPs for *OsFBN6* and its promoter regions have been derived in this study. When the haplotype results are combined with the phenotypes, the most likely functional SNPs can be selected. The haplotype analysis of the three FBN genes showed that there was a high correlation between the BS infection score and the haplotype classification, and especially *OsFBN6* showed a difference in the four haplotype infection scores was less than 0.0001, indicating that the haplotype of *OsFBN6* could be completely used for screening the BS resistant varieties. The SNP of *OsFBN6* located in Chr4_30690229 caused allelic variation from C to T in the promoter region, making haplotype 3 highly resistant with the lowest infection score among all. The mutation of Chr4_30688750 was located in the CDS region in haplotype 2 but did not let the amino acid mutate, and haplotype 3 had an advantage of the proportion in the R line over haplotype 2; therefore, the SNP of Chr4_30690229 is considered a functional site located in the *OsFBN6* promoter region. The Caps marker developed by Chr4_30690229 can be directly applied to the breeding application of screening BS-resistant rice varieties.

From the viewpoint of rice breeding, the identification of new genes related to disease susceptibility or resistance is agronomically important. There were two types of genes that confer broad-spectrum disease resistance; *BROAD-SPECTRUM RESISTANCE 1* (*BSR1*)-overexpressing lines are resistant to bacterial pathogen *Xoo* and fungal pathogens *M. oryzae* and *C. miyabeanus* [56]. *OsWAK25*, encoding one of the wall-associated kinase (WAK) subfamily members, confers resistance to hemibiotrophic pathogens but increases susceptibility to the necrotrophic fungal pathogens *C. miyabeanus* [57]. In an article on phylogenetic tree analysis of the FBN family in Arabidopsis and rice, it can be seen that the homology of FBN5 and FBN6 in rice and Arabidopsis is extremely high [58]. The chloroplast is an organelle specialized in photosynthesis and biosynthesis in plants, including phytohormones, secondary metabolites, aromatic amino acids, and fatty acids [59]. Chloroplasts in plant cells have their own DNA containing approximately 120–130 genes that drive photosynthesis in chloroplasts and are translated into proteins [60]. Several key proteins encoded by chloroplasts play vital roles in coordinating effective plant immune responses [61]. Plants have two immune systems: One is membrane-localized pattern recognition receptors that respond to pathogen-associated molecular patterns (PAMPs) via PAMP-triggered immunity. When pathogens successfully escape PAMP-triggered immunity, they secrete virulence-containing effectors that can interfere with this immunity. One of the nucleotide-binding and leucine-rich repeat (NB-LRR) proteins can be specifically recognized as an effector, triggering effector-triggered immunity, the second immune system in plants [62]. JA synthetic precursor 12-oxo-phytodienoic acid (OPDA) enantiomer, 9S,13S/cis-(+)-OPDA can accumulate in chloroplasts, and the biosynthesis of JA is regulated by the accumulation of plastoglobule-associated FBN 1–2 proteins [63,64].

In this study, we revealed that meJA treatment can enhance resistance to *C. miyabeanus*. The analysis of the JA pathway’s synthesis of the first specific enzyme AOS2 expression showed that the S line was highly upregulated (2000-fold more than CK) after *C. miyabeanus* infection, which is completely detrimental overall growth of plants, leading to a severe infection level in the S line. The expression of the S line in the group treated with meJA in advance was significantly reduced 10-fold compared with the direct fungus infection group, which may be the reason why the S line became more disease-resistant. The gene expression level of three FBNs showed different expressions regulated by *C. miyabeanus* and meJA treatment. To further verify *OsFBN6* gene function, we constructed *OsFBN6*-OE lines, which showed induced resistance against BS than NIP, indicating that *OsFBN6* positively regulated rice resistance to BS. *OsFBN6,* as a functional protein localized in chloroplasts not only related to high-light stress in *Arabidopsis* but also involved in rice BS resistance. The primary key enzyme synthesized by JA even showed great gene regulation range when infected with *C. miyabeanus*, indicating the JA pathway’s response to against BS infection in rice.

## Figures and Tables

**Figure 1 plants-13-03302-f001:**
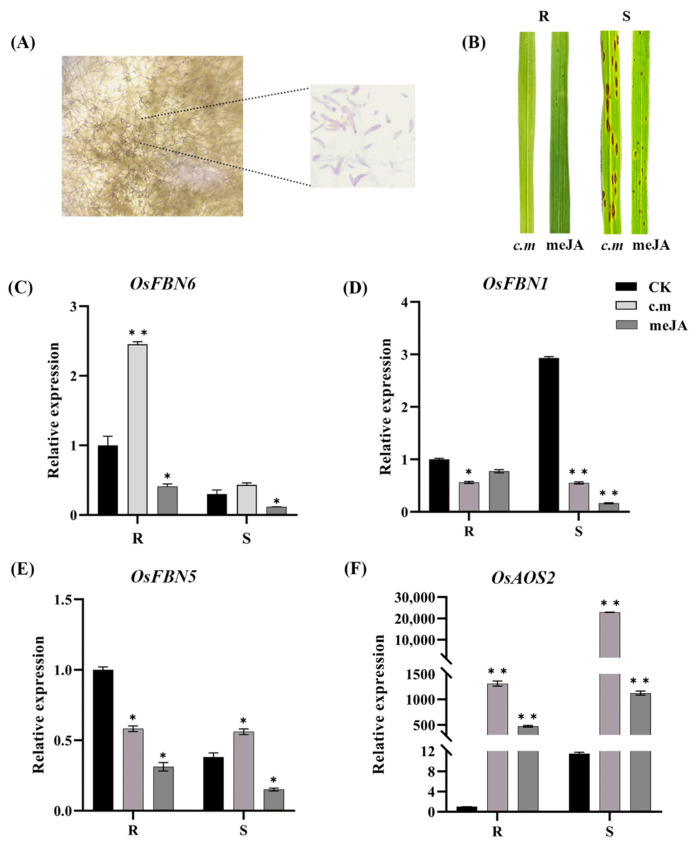
(**A**) The structure of *C. miyabeanus 36* conidia. (**B**) The phenotype of R and S lines inoculated with *C. miyabeanus* and meJA. (**C**–**F**) Reverse transcription-quantitative polymerase chain reaction (RT-qPCR) measurements of *OsFBN6*, *OsFBN1*, *OsFBN5*, and *OsAOS2* in rice leaves infected with *C. miyabeanus* and meJA. CK, control group; (*n* = 3, Student’s *t*-test; *, *p* < 0.05; **, *p* < 0.01).

**Figure 2 plants-13-03302-f002:**
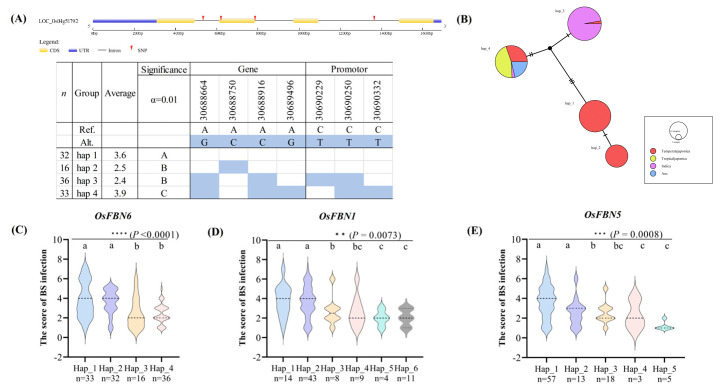
Haplotype analysis. (**A**) Schematic of gene structure and positions of SNPs in *OsFBN6*. Red arrows indicate SNP sites in the gene region. The average indicates the mean of BS infection scores of all varieties in each haplotype. The subset for α = 0.01 represents a very significant level of difference average of infection in each group. The blue squares represent mutant SNPS in each haplotype. (**B**) Haplotype network analysis in *OsFBN6*. Different colors indicate each rice subspecies, and the circle size represents the number of varieties in each haplotype. (**C**–**E**) The violin plot of BS infection score distribution for all varieties in each haplotype in *OsFBN6*, *OsFBN1* and *OsFBN5*. The dashed line represents the median, and n = the number of varieties in each haplotype. Statistical analysis finished by ANOVA and Duncan’s test, the asterisk represents the significant level at; **, *p* < 0.01; ***, *p* < 0.001; ****, *p* < 0.0001).

**Figure 3 plants-13-03302-f003:**
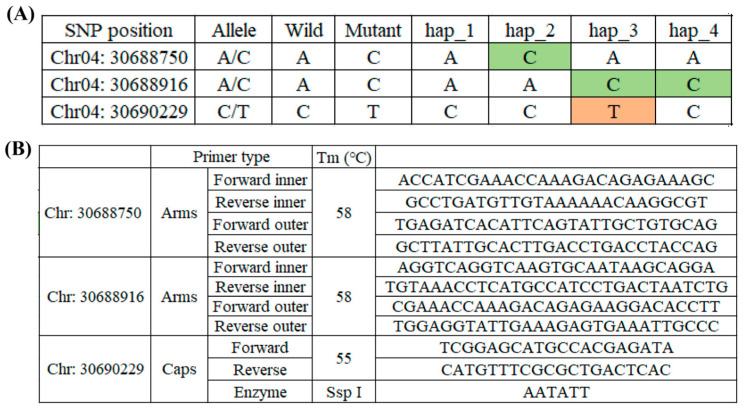
Marker designing. (**A**) Each haplotype has a different allele between wild and mutant lines in three SNP positions. Mutant alleles are color-coded. (**B**) The primer list of the markers: Arms primer has two pairs, and Caps primer has 1 pair with an SspI-cutting site.

**Figure 4 plants-13-03302-f004:**
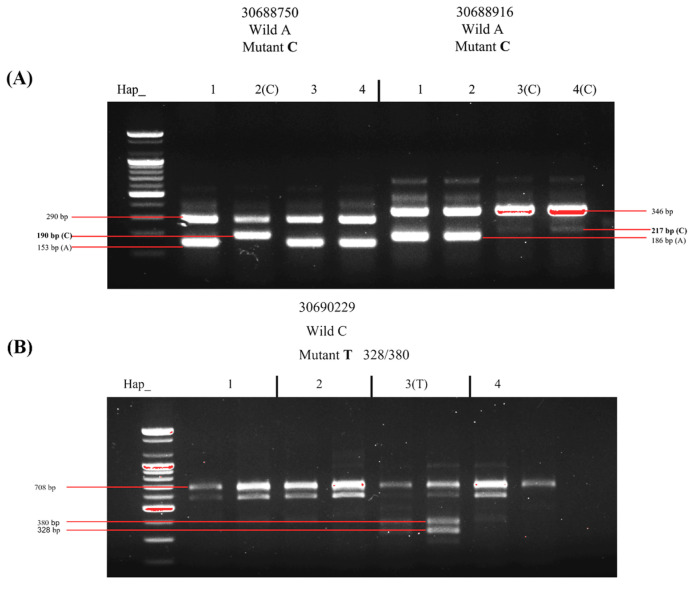
Results of agarose gel electrophoresis. (**A**) The DNA fragments were amplified using Arms primers. (**B**) The DNA fragments were amplified using Caps primers. Mutant alleles are bold numbers.

**Figure 5 plants-13-03302-f005:**
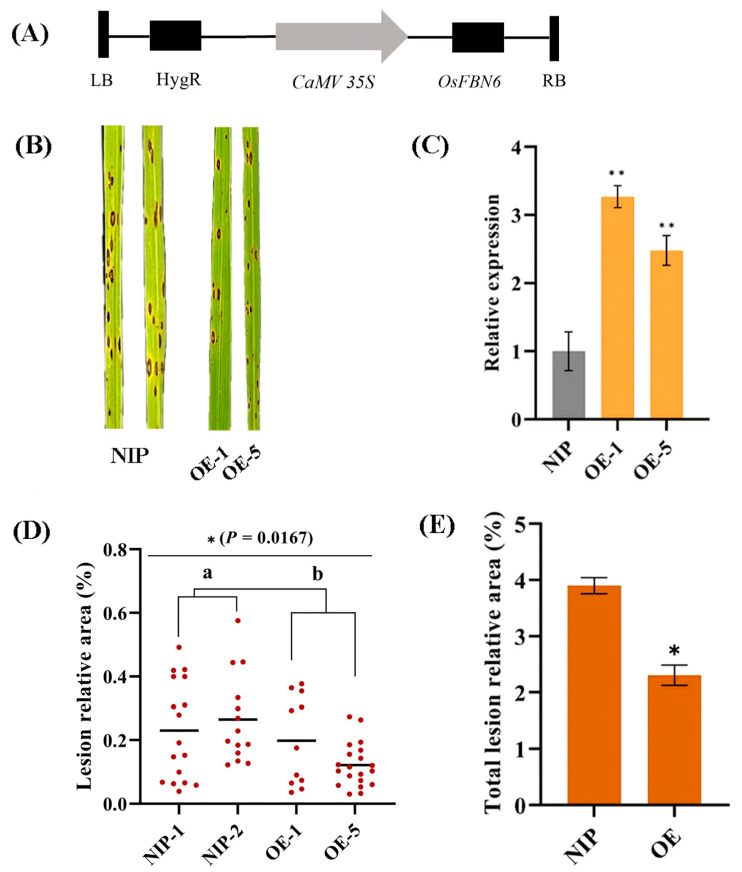
Functional identification of the *OsFBN6* overexpression lines. (**A**) Structure and schematic diagram of the plasmid. (**B**) WT and OE line phenotypes infected with *C. miyabeanus*. (**C**) The reverse transcription-quantitative polymerase chain reaction (RT-qPCR) measurements of the WT and OE line gene expressions in rice leaves infected with *C. miyabeanus.* (**D**) Lesion-relative area and amounts of WT and OE lines inoculated with *C. miyabeanus*. The value of the red dot represents the relative proportion of the leaf area of each spot; the number represents the number of spots, and the bars represent the average of all spots. Statistical analysis finished by ANOVA and Duncan’s test, the asterisk represents the significant level at *, *p* < 0.05. (**E**) The total lesion relative area of infection with *C. miyabeanus* (*n* = 3, Student’s *t*-test; *, *p* < 0.05; **, *p* < 0.01).

## Data Availability

The data sets used and/or analyzed during the current study are available from the corresponding author upon reasonable request.

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
