# Peer review of "OsFBN6 Enhances Brown Spot Disease Resistance in Rice"

_plants, 2024, doi:10.3390/plants13233302_

Round 1

Reviewer 1 Report

Comments and Suggestions for Authors

I have the following minor comments need to be addressed:

(1) Page 8 line 278: "The subcellular localization of OsFBN6 was investigated by transiently expressing the construct in NIP via E. coli". The method here should be PEG- mediated transient expression according to the method part. E.coli might be wrong. please revised that.

 (2) Page 8 line 278-280: OsFBN6 protein signal was localized in the chloroplast. Since no chloroplast marker is used in this figure, it is not convincing to show OsFBN6 is localized in chloroplast.

(3) For Figure 5 D, there is no statistical analysis to show a significant difference.

Author Response

Comments 1: Page 8 line 278: "The subcellular localization of OsFBN6 was investigated by transiently expressing the construct in NIP via E. coli". The method here should be PEG- mediated transient expression according to the method part. E.coli might be wrong. please revised that.

Response 1: Thank you for your pointing this out. We agree with this comment. Chloroplast marker should be used as a marker in OsFBN6 subcellular localization analysis proposed in comments 2, and previous studies have already shown that the rice OsFBN6 gene is located in chloroplasts (citation No. 27), so we decided to delete the experimental results on OsFBN6 subcellular localization.

Comments 2: Page 8 line 278-280: OsFBN6 protein signal was localized in the chloroplast. Since no chloroplast marker is used in this figure, it is not convincing to show OsFBN6 is localized in chloroplast.

Response 2: Thank you for your pointing this out. We agree with this comment. Since the results of phylogenetic tree analysis of FBN family in rice and Arabidopsis have been published, it is clear that the FBN family is all located in chloroplasts (quotation is as follows). Therefore, we think this part does not significantly impact our main conclusions.

However, as it is challenging to conduct the additional experiments suggested within 10 days, we propose to delete this section in this study. We would be happy to follow your further recommendations on this matter if you have additional opinions.

Kim I & Kim HU (2022) The mysterious role of fibrillin in plastid metabolism: current advances in understanding. Journal of experimental botany 73:2751-2764.
: This review discusses the discovery and functional roles of FBNs in various plastids, emphasizing their presence in chloroplasts and their involvement in plastid stability and metabolism.

Li, J et al. (2020) Phylogeny, structural diversity and genome-wide expression analysis of fibrillin family genes in rice. Phytochemistry 175: 112377.
: This study investigates the evolutionary relationships, structural diversity, and expression patterns of FBN genes in rice, highlighting their role in chloroplast function, stress responses, and metabolic processes.

Ariadna IS et al. (2024) Arabidopsis FIBRILLIN6 influences carotenoid biosynthesis by directly promoting phytoene synthase activity, Plant Physiology 194: 1662–1673
: This study highlights the role of FBN6 in chloroplasts, demonstrating its interaction with phytoene synthase and its impact on carotenoid biosynthesis.

Choi YR et al. (2021) Chloroplast Localized FIBRILLIN11 Is Involved in the Osmotic Stress Response during Arabidopsis Seed Germination. Biology 10:368.
: This research identifies FBN11's localization in chloroplasts and its role in mediating osmotic stress tolerance during seed germination.

El-Sappah AH et al. (2024) Fibrillin gene family and its role in plant growth, development, and abiotic stress. Front. Plant Sci. 15:1453974.
: This mini-review delves into the structural attributes, phylogenetic classification, and functional roles of FBNs in plants, with a special focus on their effectiveness in mitigating abiotic stresses, particularly drought stress.

Kim I et al. (2022) Fibrillin2 in chloroplast plastoglobules participates in photoprotection and jasmonate-induced senescence, Plant Physiology 189:1363–1379
: This study explores the role of FBN2 in chloroplast plastoglobules, highlighting its involvement in photoprotection and jasmonate-induced senescence.

Comments 3:  For Figure 5 D, there is no statistical analysis to show a significant difference.

Response 3: Thank you for your pointing this out. We agree with this comment. We added the t-test between wildtype and overexpression lines in Figure 5 D, and the Duncan test in each Haps was also added in Figure 2 C-E.

Reviewer 2 Report

Comments and Suggestions for Authors

Please refer to the following suggestions for the improvement.

From the results of this paper, it seems difficult to deduce that FBN6 promotes the production of OPDA, a JA precursor, in a conjugate manner with AOS. As the authors claim, I understand that FBN6 is involved in increased resistance to BS based on the results of haplotype analysis and experiments of overexpression lines. However, there seems to be a lack of direct evidence that FBN6 enhances the immune system against BS through JA signaling. In particular, the expression of the OsFBN6 gene is lower than that in untreated rice in sensitive rice variety treated with JA and Cm (Fig 1C). I don't think this is direct evidence for that FBN6 confers resistance against BS through JA signaling.

Although the authors investigated the subcellular phenomena, they used the whole rice leaf blades for RNA extraction. To analyze this kind of phenomena, it is necessary to survey the small site in which infection actually occurs (the leaf blades used for RNA extraction in this study seem to contain a mixture of lesions and healthy parts). Is there any problem in discussing subcellular phenomena using such bulk extractions?

The picture in Figure 6 is difficult to understand. In particular, JA synthetic cascade in the chloroplast is difficult to understand. please improve it. I think that JA signaling is located downstream of OPDA synthesis.

Line 124-138

Please describe clearly the normalizer used in the relative quantitative RT-PCR performed for gene expression analysis.

In Figure 3 (A), the columns of Hap3 and Hap4 are opposite. Please fix it.

That's all.

Author Response

Comments 1: From the results of this paper, it seems difficult to deduce that FBN6 promotes the production of OPDA, a JA precursor, in a conjugate manner with AOS. As the authors claim, I understand that FBN6 is involved in increased resistance to BS based on the results of haplotype analysis and experiments of overexpression lines. However, there seems to be a lack of direct evidence that FBN6 enhances the immune system against BS through JA signaling. In particular, the expression of the OsFBN6 gene is lower than that in untreated rice in sensitive rice variety treated with JA and Cm (Fig 1C). I don't think this is direct evidence for that FBN6 confers resistance against BS through JA signaling.

Response 1: Thank you for your pointing this out. We agree with this comment. Indeed, in this study, it can be seen that JA treatment can significantly improve the resistance of rice to BS, and BS infection has a huge impact on the expression of AOS2, a key enzyme in the JA pathway, which can only prove that the JA pathway does respond to the resistance of rice to BS disease.

The functional study of OsFBN6 gene also proved that the gene played a positive regulatory role in the resistance of rice to BS. These two results do not prove that FBN6 is involved in the JA pathway response when rice is resistant to BS.

Therefore, we have revised the abstract and the discussion sections about the relationship between OsFBN6 and JA pathway to only elaborate on the conclusions obtained in this study.

Comments 2: Although the authors investigated the subcellular phenomena, they used the whole rice leaf blades for RNA extraction. To analyze this kind of phenomena, it is necessary to survey the small site in which infection actually occurs (the leaf blades used for RNA extraction in this study seem to contain a mixture of lesions and healthy parts). Is there any problem in discussing subcellular phenomena using such bulk extractions?

Response 2: Thank you for your pointing this out. We agree with this comment. Since the results of phylogenetic tree analysis of FBN family in rice and Arabidopsis have been published, it is clear that the FBN family is all located in chloroplasts (quotation is as follows), so the relevant experiments and results on subcellular localization of OsFBN6 were deleted in this study.

Kim, Inyoung, and Hyun Uk Kim. "The mysterious role of fibrillin in plastid metabolism: current advances in understanding." Journal of experimental botany 73.9 (2022): 2751-2764.

Li, Jiajia, et al. "Phylogeny, structural diversity and genome-wide expression analysis of fibrillin family genes in rice." Phytochemistry 175 (2020): 112377.

Comments 3: The picture in Figure 6 is difficult to understand. In particular, JA synthetic cascade in the chloroplast is difficult to understand. please improve it. I think that JA signaling is located downstream of OPDA synthesis.

Response 3: Thank you for your pointing this out. We agree with this comment. It is true that the hypothesis in Figure 6 cannot be completely derived from the existing research results, so we deleted the hypothesis in Figure 6. In future studies, we will use more experimental results to verify this hypothesis.

Comments 4: Line 124-138, Please describe clearly the normalizer used in the relative quantitative RT-PCR performed for gene expression analysis.

Response 4: Thank you for your pointing this out. We agree with this comment. We included normalizer in the relative quantitative RT-PCR performed for gene expression analysis. “The cycle number threshold (Ct value) was employed to calculate relative mRNA expression levels. The Ct value for each target gene was normalized by subtracting the Ct value of the rice (ACTIN 1, LOC_Os03g50885) gene.”

Comments 5: In Figure 3 (A), the columns of Hap3 and Hap4 are opposite. Please fix it.

Response 5: Thank you for your pointing this out. We agree with this comment. We modified this mistake in Figure 3 (A).

Round 2

Reviewer 2 Report

Comments and Suggestions for Authors

The parts I pointed out have been improved.

I recommend that this paper would be published in journal Plants.

Author Response

Comment: The parts I pointed out have been improved.

I recommend that this paper would be published in journal Plants.

Response: Thank you for your recognition of this research.